

# Compacting the Description of a Time-Dependent Multivariable System and Its Time-Dependent Multivariable Driver by Reducing the System and Driver State Vectors to Aggregate Scalars: The Earth's Solar-Wind-Driven Magnetosphere

Joseph E. Borovsky[1] and Adnane Osmane[2]

[1]Center for Space Plasma Physics, Space Science Institute, Boulder, Colorado, USA
[2]Department of Physics, University of Helsinki, Helsinki, Finland

*Correspondence to*: Joseph E. Borovsky (jborovsky@spacescience.org)

**Abstract.** Using the solar-wind-driven magnetosphere-ionosphere-thermosphere system, a methodology is developed to
reduce a state-vector description of a time-dependent driven system to a composite scalar picture of the activity in the
system. The technique uses canonical correlation analysis to reduce the time-dependent system and driver state vectors to
time-dependent system and driver scalars, with the scalars describing the response in the system that is most-closely related
to the driver. This reduced description has advantages: low noise, high prediction efficiency, linearity in the described
system response to the driver, and compactness. The methodology identifies independent modes of reaction of a system to its
driver. The analysis of the magnetospheric system is demonstrated. Using autocorrelation analysis, Jensen-Shannon
complexity analysis, and permutation-entropy analysis the properties of the derived aggregate scalars are assessed. This
state-vector-reduction technique may be useful for other multivariable systems driven by multiple inputs.

## 1. Introduction

20       In this report a methodology is described that can produce a compact description of the behavior of a time-dependent, multivariable system driven by a time-dependent, multivariable driver or by multiple drivers. The diverse variables describing the system may be intercorrelated, and the variables describing the driving may be intercorrelated. The methodology was developed to gain an understanding of the Earth's magnetosphere-ionosphere-thermosphere system as driven by the solar wind. To utilize the methodology the system and its driver are conceptualized by a time-dependent,
multidimensional system state vector $\underline{S}(t)$ and a time-dependent, multidimensional driver state vector $\underline{D}(t)$, with the assumption that the driver vector $\underline{D}$ affects the system vector $\underline{S}$, but not vice versa, written $\underline{D} \rightarrow \underline{S}$. The individual time-dependent scalar variables making up the state vector $\underline{S}(t)$ are time-dependent measures of various forms of activity in the system and various properties of the system and the individual time-dependent scalar variables making up the driver state vector $\underline{D}(t)$ are various time-dependent measures of the properties of the drivers of the system. We will utilize the correlation
properties between the components (individual time-dependent variables) of $\underline{S}$ and the components of $\underline{D}$. Canonical correlation analysis (CCA) will be used to derive scalar projections (dot products) of $\underline{S}(t)$ and scalar projections of $\underline{D}(t)$ that have the highest Pearson linear correlation coefficient between them. The derived scalar projections $S_{(1)}(t)$, $S_{(2)}(t)$, $S_{(3)}(t)$, … of the vector $\underline{S}(t)$ will be composite (aggregate) measures of activity in the system and the derived scalar projections $D_{(1)}(t)$, $D_{(2)}(t)$, $D_{(3)}(t)$, … of the vector $\underline{D}(t)$ will be the composite drivers of $S_{(1)}(t)$, $S_{(2)}(t)$, $S_{(3)}(t)$, …., respectively. This reduced
scalar picture $D_{(i)} \rightarrow S_{(i)}$ of the system driven by the driver focuses on the time-dependent properties of the system that react to the driver. By maximizing the correlations, the predictability of the system from a knowledge of the state of the driver is also maximized.

        The system used to develop this methodology is the Earth's magnetosphere-ionosphere-thermosphere system driven by the time-dependent solar wind. The spatial domain wherein the Earth's magnetic field dominates over the solar wind is
known as the magnetosphere. The interaction between the solar wind and the magnetosphere is surprisingly complex and the



magnetosphere's evolution in response to the time-varying solar wind is rich and diverse. The magnetospheric system is characterized by multiple subsystems that interact with each other (cf. Lyon, 2000; Otto, 2005; Siscoe, 2011; Eastwood et al., 2015; Borovsky and Valdivia, 2018): almost 6 orders of magnitude of spatial scales are involved in the global behavior of the magnetosphere, from ~1 km to ~$6\times10^5$ km. This system is highly coupled, dynamic, with memory and with feedback

loops. Multiple physical processes act to couple the various subsystems, with the strength of the couplings evolving with time as the subsystems evolve owing to the couplings. Even after a half of a century of measurements and analysis, its subsystems and the couplings between its subsystems are not fully understood (Stern, 1989, 1996; Denton et al., 2016). It has been argued that the system adjectives "adaptive", "nonlinear", "dissipative", and "complex" apply to the magnetospheric system (Borovsky and Valdivia, 2018). (See also the earlier systems analyses by Horton et al. (1999),

Chapman et al. (2004), Valdivia et al. (2005, 2013), and Sharma (2010)). The magnetospheric system is well measured: there are hundreds of thousands of hours of simultaneous measurements of various aspects of the magnetospheric system and its solar-wind driver over the five decades of the "space age" (cf. Stern, 1989, 1996; King and Papitashvili, 2005).

The solar-wind-driven magnetospheric system very cleanly follows the $\underline{D}\rightarrow\underline{S}$ picture where the driver affects the system, but the system does not affect the driver. The Earth's magnetosphere has no influence whatsoever on the properties

of the solar wind that passes the Earth. Measurements of this magnetospheric system will be used in Sections 2 and 3 to explore the mathematical reduction of the state-vector $\underline{D}(t)\rightarrow\underline{S}(t)$ picture to the composite-scalar $D_{(i)}(t)\rightarrow S_{(i)}(t)$ picture. Table 1 lists the 9 time-dependent measurements of the magnetosphere in the system state vector $\underline{S}$ and the 8 time-dependent measurements of the solar wind in the driver state vector $\underline{D}$. The individual variables in the system state vector and in the driver state vector are described in the Appendix.

This report is organized as follows. In Section 2 the CCA approach is applied to the magnetospheric system driven by the solar wind to derive the first three time-dependent sets of composite variables $S_{(1)}(t)$ and $D_{(1)}(t)$, $S_{(2)}(t)$ and $D_{(2)}(t)$, and $S_{(3)}(t)$ and $D_{(3)}(t)$. from the state vectors $\underline{S}(t)$ and $\underline{D}(t)$. In Section 3 the three sets of composite variables $S_{(i)}$ and $D_{(i)}$ for the magnetospheric system are explored and the complexity-entropy properties of the aggregate variable $S_{(1)}(t)$ are analyzed. In Section 4 the advantages of the reduced $D_{(i)}\rightarrow S_{(i)}$ scalar description are examined: these advantages include (a) a compact

description of global system-wide reactions to variations in the driver, (b) increased predictability of the system from a knowledge of the driver, (c) linearity in the description of the system's response to the driver, and (d) lower noise in correlations between the system variables and the driver variables. The reduced scalar picture can also reveal independent modes of reaction of the system to the driver, providing insight into the behavior of the system in reaction to complexities in the driver. The variables of the magnetospheric and solar-wind state vectors are described in the Appendix.



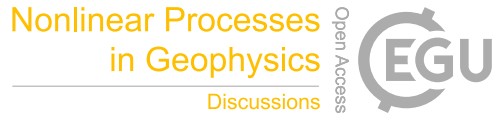

**2. Creation of Composite (Aggregate) Variables from the State Vectors**

Using the Earth's magnetosphere-ionosphere-thermosphere system as driven by the solar wind, the reduction of a time-dependent state-vector picture $\underline{D}(t) \rightarrow \underline{S}(t)$ to the time-dependent composite-variable-pair picture $D_{(i)}(t) \rightarrow S_{(i)}(t)$ will be performed. The 9 measured variables chosen for the 9-dimensional magnetospheric system state vector $\underline{S}$ appear in the first

column of Table 1, and the 8 measured variables chosen for the 8-dimensional solar-wind driver state vector $\underline{D}$ appear in the second column of Table 1, with explanations of those measures deferred to the Appendix.

One-hour averages of all magnetospheric and solar-wind variables are used in the years 1991-2007. No time lags are used between the solar-wind measurements and the magnetospheric measurements: most expected time lags will be about 1-hr (e.g. Clauer et al., 1981; Smith et al., 1999), which is the time resolution of the data set.

Canonical correlation analysis (CCA) is applied to the time-dependent state vectors $\underline{S}(t)$ and $\underline{D}(t)$. CCA finds correlation patterns between two multivariable data sets (Nimon et al., 2010; Hair et al., 2010). It yields pairs of composite (aggregate) variables (a) that are linear combinations of the variables of the two data sets and (b) that have maximal correlations with each other. Each pair of composite variables is called the "Nth canonical correlation". From the data sets of $\underline{S}(t)$ and $\underline{D}(t)$ the first pair of composite variables yielded (the first canonical variates) are $S_{(1)}(t)$ and $D_{(1)}(t)$: these two

variables are projections of $\underline{S}$ and $\underline{D}$ given by $S_{(1)}(t) = \underline{C}_{S1} \bullet \underline{S}(t)$ and $D_{(1)}(t) = \underline{C}_{D1} \bullet \underline{D}(t)$ where $\underline{C}_{S1}$ and $\underline{C}_{D1}$ are time-independent coefficient (weight) vectors. $S_{(1)}$ and $D_{(1)}$ are the composite variables from $\underline{S}$ and $\underline{D}$ that have the highest Pearson linear correlation coefficient with each other. Here, CCA is in a sense creating the system function $S_{(1)}(t)$ that is most reactive to the driver vector $\underline{D}(t)$ and creating the driver scalar function $D_{(1)}(t)$ that describes that driving. CCA then yields other pairs of composite variables $S_{(2)}$ and $D_{(2)}$ (the second canonical correlation), $S_{(3)}$ and $D_{(3)}$ (the third canonical

correlation), etc. $S_{(2)}$ and $D_{(2)}$ are the projections of $\underline{S}$ and $\underline{D}$ that have the highest correlation with each other, providing that $S_{(2)}$ and $D_{(2)}$ are uncorrelated with $S_{(1)}$ and $D_{(1)}$. $S_{(3)}$ and $D_{(3)}$ are the projections of $\underline{S}$ and $\underline{D}$ that have the highest correlations with each other, provided they are uncorrelated with $S_{(1)}$, $S_{(2)}$, $D_{(1)}$, and $D_{(2)}$. $S_{(1)}(t)$, $S_{(2)}(t)$, and $S_{(3)}(t)$ represent three independent modes of reaction of the global system to the driver $\underline{D}(t)$. The CCA process will identify these modes (and their respective drivers).

CCA is a matrix equation solution, non-iterative, that yields a single unique solution (*Johnson and Wichern*, 2007). CCA operates on standardized variables (with the mean value subtracted and the values then divided by the standard deviation), denoted with an asterisk *. (For each variable the mean value and standard deviation are calculated for the entire data set.) CCA operates most efficiently on variables that are Gaussian distributed: hence the logarithms of some variables are used to yield more-Gaussian-like distributions. All standardized variables v* have a mean value of zero, a standard deviation

deviation unity, and no units.

When CCA is applied to the 1991-2007 $\underline{S}(t)$ and $\underline{D}(t)$ data sets (see Table 1), the first canonical pair of time-dependent variables is

$$S_{(1)} = 0.0260 \log_{10}(1 + |AL|)^* + 0.1151 \log_{10}(1 + |AU|)^* + 0.2160 |PCI|^*$$
$$+ 0.1451 Kp^* + 0.2881 \log_{10}(1 + am)^* + 0.0201 d|Dst|/dt^*$$
$$+ 0.0492 \log_{10}(0.01 + mP_e)^* + 0.2531 \log_{10}(0.01 + mP_i)^*$$
$$+ 0.0854 \log_{10}(0.01 + P_{ips})^* \tag{1a}$$
$$D_{(1)} = 0.8378 \log_{10}(v_{sw})^* + 0.6876 \log_{10}(n_{sw})^* + 0.1018 \log_{10}(F_{10.7})^*$$
$$- 0.1676 (-B_z)^* + 0.3547 f(M)^* + 0.3844 <\sin^2(\theta_{clock}/2)>_3^*$$
$$+ 0.0960 <\theta_{Bn}>_3^* + 0.0943 \log_{10}(0.1 + |\Delta B|)^* \quad . \tag{1b}$$

$S_{(1)}$ and $D_{(1)}$ have mean values of zero and standard deviations of unity. The derived composite variables given by expressions (1) are robust and reproducible: applying the CCA process to various subsets of the full 1991-2007 data set, the CCA process repeatedly yields essentially the same coefficients that are in expressions (1a) and (1b) (cf. Borovsky and Denton, 2018).



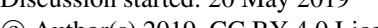


CCA applied to the time-dependent state vectors $\underline{S}(t)$ and $\underline{D}(t)$ for the 1991-2007 data set yields the second canonical pair of time-dependent scalar variables as

$$S_{(2)} = -0.2628 \log_{10}(1 + |AL|)^* - 0.0874 \log_{10}(1 + |AU|)^* - 0.1302 |PCI|^*$$
$$- 0.0556 \, Kp^* + 0.1928 \log_{10}(1 + am)^* + 0.0028 \, d|Dst|/dt^*$$
5
$$- 0.8506 \log_{10}(0.01 + mP_e)^* + 0.9218 \log_{10}(0.01 + mP_i)^*$$
$$+ 0.3493 \log_{10}(0.01 + P_{ips})^* \tag{2a}$$
$$D_{(2)} = 0.1195 \log_{10}(v_{sw})^* + 0.8874 \log_{10}(n_{sw})^* + 0.1202 \log_{10}(F_{10.7})^*$$
$$- 0.1138 \, (-B_z)^* + 0.2669 \, f(M)^* - 0.5079 <\sin^2(\theta_{clock}/2)>_3^*$$
$$- 0.0186 <\theta_{Bn}>_3^* + 0.0260 \log_{10}(0.1 + |\Delta B|)^* \quad . \tag{2b}$$

10 and for the 1991-2007 data set CCA yields the third canonical pair of time-dependent scalar variables as

$$S_{(3)} = -0.1796 \log_{10}(1 + |AL|)^* - 0.2220 \log_{10}(1 + |AU|)^* - 1.0351 |PCI|^*$$
$$+ 0.8265 \, Kp^* + 0.5809 \log_{10}(1 + am)^* - 0.2169 \, d|Dst|/dt^*$$
$$+ 0.3856 \log_{10}(0.01 + mP_e)^* - 0.6100 \log_{10}(0.01 + mP_i)^*$$
$$+ 0.1064 \log_{10}(0.01 + P_{ips})^* \tag{3a}$$
15
$$D_{(3)} = 0.4241 \log_{10}(v_{sw})^* - 0.1985 \log_{10}(n_{sw})^* - 0.1404 \log_{10}(F_{10.7})^*$$
$$-0.6704 \, Z(-B_z)^* - 0.1008 \, f(M)^* + 0.0572 <\sin^2(\theta_{clock}/2)>_3^*$$
$$- 0.3134 <\theta_{Bn}>_3^* + 0.3055 \log_{10}(0.1 + |\Delta B|)^* \quad . \tag{3b}$$

The properties of $S_{(1)}(t)$ and $D_{(1)}(t)$, $S_{(2)}(t)$ and $D_{(2)}(t)$, and $S_{(3)}(t)$ and $D_{(3)}(t)$ as given by expressions (1) - (3) are explored in Section 3.

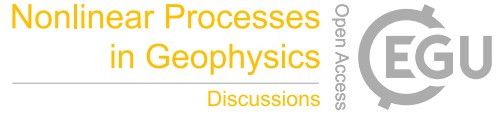

### 3. Properties of the Scalar Reduced Picture for the Magnetospheric System

The three sets of composite variables $S_{(1)}$ and $D_{(1)}$, $S_{(2)}$ and $D_{(2)}$, and $S_{(3)}$ and $D_{(3)}$ for the magnetospheric system are explored and the advantages of the reduced $D_{(i)} \rightarrow S_{(i)}$ scalar description are investigated.

**3.1. The Primary Mode of System Response as Represented by $D_{(1)} \rightarrow S_{(1)}$**

In Figure 1 the composite system variable $S_{(1)}$ (as given by expression (1a)) is plotted for the years 1991-2007 as a function of the composite driver variable $D_{(1)}$ (as given by expression (1b)). Each black point in Figure 1 represents 1 hour of data. The Pearson linear correlation coefficient between $S_{(1)}$ and $D_{(1)}$ for the 1991-2007 data set is $r_{corr} = 0.921$. Accordingly, $r_{corr}^2$ = 84.8% of the variance of the system function $S_{(1)}(t)$ is described by the driver function $D_{(1)}(t)$, and so 15.2% of the variance

of $S_{(1)}$ is unaccounted for by $D_{(1)}$. The blue line in Figure 1 is a linear-regression fit to $S_{(1)}$ and the red curve is a 50-point vertical running average of the black points. Note the approximate linearity of system variable $S_{(1)}$ as a function of driver variable $D_{(1)}$, indicated by the manner in which the running average tracks the linear-regression line.

Note that whereas the correlation coefficient between $S_{(1)}(t)$ and $D_{(1)}(t)$ is $r_{corr} = 0.921$, the maximum correlation coefficient between any single variable in the system state vector $\underline{S}(t)$ and any single variable in the driver state vector $\underline{D}(t)$ is

only $r_{corr} = 0.586$ (between $<\sin^2(\theta_{clock}/2)>_3$ and $\log_{10}(1 + |AL|)$).

In the six panels of Figure 2 the coefficients of the six vectors $C_{S1}$, $C_{D1}$, $C_{S2}$, $C_{D2}$, $C_{S3}$, and $C_{D3}$ are plotted. (These are the coefficients in expressions (1) - (3).) Examining these six panels enables the reaction modes represented by $S_{(1)}$, $S_{(2)}$, and $S_{(3)}$ to be interpreted as well as their drivers $D_{(1)}$, $D_{(2)}$, and $D_{(3)}$. Figure 2a indicates that all coefficients of $S_{(1)}$ are positive: this indicates a mode of the magnetospheric system in which all measures of activity in the system vector $\underline{S}$

increase or decrease in unison, with $S_{(1)}$ representing a "global activity index". Figure 2b indicates that all of the coefficients of $D_{(1)}$ are positive. The variables in the driver state vector $\underline{D}$ (Table 1) and their signs were all chosen so that a positive increase in each variable would result in a generally accepted increase in magnetospheric activity. The individual variables on the right-hand side of expression (1b) have all been correlatively associated with the driving of magnetospheric activity (Berthelier, 1976; Borovsky and Funsten, 2003; Newell et al., 2007; Borovsky and Denton, 2014; Borovsky and Birn, 2014;

Osmane et al., 2015). $S_{(1)}$ is selected by the CCA process to have highest correlation with solar-wind variability: $S_{(1)}$ is focused on activity that reacts to the solar-wind driver.

Using the linear-regression curve in Figure 1 as a "prediction" of the value of $S_{(1)}$ from a knowledge of the value of $D_{(1)}$ yields

$$S_{(1)pred} = 0.9209\, D_{(1)} - 4.4 \times 10^{-5} \qquad . \qquad\qquad (4)$$

In Figure 3 the autocorrelation functions of $S_{(1)}(t)$ (red curve), $D_{(1)}(t)$ (blue curve), and $S_{(1)}(t)-S_{(1)pred}(t)$ (green curve) are plotted. In Figure 3a it is seen that the autocorrelation functions of $S_{(1)}$ and $D_{(1)}$ are very similar, with 1/e autocorrelation times of 23.3 hr for $S_{(1)}$ and 22.7 hr for $D_{(1)}$. In Figure 3b the three autocorrelation functions are plotted for time shifts up to 40 days. Note the 27-day peak in the autocorrelation functions of $D_{(1)}(t)$ and $S_{(1)}(t)$: this is associated with the 27-day rotation period of the Sun as viewed from the Earth and the persistence of features on the solar surface that give rise to solar wind

with characteristic properties. This causes the driver $\underline{D}(t)$ properties to have a 27-day periodicity, which drives the system $\underline{S}(t)$ with a 27-day periodicity.

The quantity $S_{(1)}-S_{(1)pred}$ is the portion of $S_{(1)}(t)$ that is not accounted for by $D_{(1)}(t)$, i.e., the unaccounted for variance of $S_{(1)}(t)$. $S_{(1)}(t)-S_{(1)pred}(t)$ is completely uncorrelated with $D_{(1)}(t)$. Further, $S_{(1)}(t)-S_{(1)pred}(t)$ is completely uncorrelated with each of the 8 individual solar-wind variables on the right-hand side of expression (1b). Since $S_{(1)}$ is so similar to $D_{(1)}$, the

standard analyses of the $S_{(1)}(t)$ time series (e.g. determining the correlation dimension, examining the state space, or Fourier analyzing (Sharma et al., 2005a; Vassiliadis, 2006)) would largely be an analysis of the properties of the solar-wind time series $D_{(1)}(t)$. Not so for $S_{(1)}(t)-S_{(1)pred}(t)$, which is uncorrelated with $D_{(1)}$. The autocorrelation function of $S_{(1)}(t)-S_{(1)pred}(t)$ in Figure 3a is very different from the autocorrelation function of $D_{(1)}$: the 1/e autocorrelation time of $S_{(1)}(t)-S_{(1)pred}(t)$ is 2.4 hr.



Determining what the unaccounted-for variance $S_{(1)}$-$S_{(1)pred}$ originates from is of great interest. Four suggestions of what contributes to $S_{(1)}$-$S_{(1)pred}$ are made here. First, some fraction of $S_{(1)}$-$S_{(1)pred}$ may be associated with noise in the various measurements of the magnetospheric system and of the solar wind. Shot noise (random noise in the values of the variables) would have an autocorrelation time of less than 1 hr, the autocorrelation function of the shot-noise going from 1 to 0 in one

data-resolution time shift (cf. Sect. 2.4 of Borovsky et al. (1997)). Second, some fraction of $S_{(1)}(t)$-$S_{(1)pred}(t)$ may be owed to errors in the measurement values in the state vectors $\underline{S}(t)$ and $\underline{D}(t)$. Errors in the values of the variables of $\underline{D}$ could be caused by the spatial structure of the solar wind and the measuring spacecraft upstream of the Earth not intercepting the exact solar-wind structures that hit and drive the Earth (cf. Weimer et al., 2003; Borovsky, 2018): this could affect all of the variables of $\underline{D}$. Extrapolating local measures to estimate global properties can also lead to errors: this might affect the hemispheric

particle-precipitation variables $mP_e$ and $mP_i$ (Emery et al., 2008) in $\underline{S}$ and also the magnetospheric pressure values $P_{ips}$ (Borovsky, 2017) in $\underline{S}$. Variables reacting to more than one physical process (such as d|Dst|/dt and $P_{ips}$) could also appear to have error in the values when relating the values to $D_{(1)}$. Third, unaccounted-for time lags between solar-wind variables and magnetospheric variables may be resulting in weakened correlations: most time lags are 1 hr or less, but measurements of magnetospheric particle populations can have lags of several hours (e.g. Denton and Borovsky, 2009; Borovsky, 2017). The

fourth suggestion is that some fraction of $S_{(1)}(t)$-$S_{(1)pred}(t)$ might be associated with system variations that are not directly associated with the solar-wind driver as measured by $\underline{D}$. The autocorrelation time of $S_{(1)}(t)$-$S_{(1)pred}(t)$ is approximately the 2-3 hr time duration of a magnetospheric substorm (Borovsky et al., 1993; Weimer, 1994; Chu et al., 2015). Substorms are large transients in the reaction of the magnetospheric system to solar-wind driving. (Substorms have been described as self-organized criticality events in the driven magnetospheric system (Klimas et al., 2000).) The occurrence of a substorm is

notoriously difficult to predict from solar wind data (Freeman and Morley, 2004; Hsu and McPherron, 2009; Newell and Liou, 2011). The timing of substorm occurrence would be particularly difficult to infer from the 1-hr-resoluton variables going into $\underline{D}$ because of the 3-hr smoothing used on the clock-angle term $<\sin^2(\theta_{clock}/2)>_3$ in expression (1b) for $D_{(1)}$, with the clock angle being critical for substorm occurrence (Newell and Liou, 2011). The occurrence of a substorm would produce signatures in many of the variables used in $S_{(1)}$, typically an enhancement in the variable's amplitude lasting 2-3

hours (Weimer, 1994).

To investigate this substorm hypothesis for $S_{(1)}(t)$-$S_{(1)pred}(t)$, the variables $D_{(1)}(t)$, $S_{(1)}(t)$, and $S_{(1)}(t)$-$S_{(1)pred}(t)$ are superposed-epoch averaged in Figure 4 for a collection of 2155 substorm events; the collection is from Borovsky and Yakymenko (2017). The zero epoch in Figure 4 is the onset time of each of the 2155 substorms. Substorms are associated with intervals of driving of the magnetosphere (e.g. Caan et al., 1977; Morley and Freeman, 2007); this is indicated by the

increase in the superposed average of $D_{(1)}$ beginning prior to the onset time in Figure 4. However, substorms also represent a transient release of stored energy in the magnetosphere (Birn et al., 2006); this is indicated in Figure 4 by the superposed average of $S_{(1)}$ exceeding the superposed average of $D_{(1)}$ after the substorm onset and by the positive perturbation of the $S_{(1)}$-$S_{(1)pred}$ curve after onset. The $S_{(1)}$-$S_{(1)pred}$ curve indicates a transient in $S_{(1)}$ that is unaccounted for by $D_{(1)}$ associated with the occurrence of substorms. The autocorrelation time of the green superposed-average $S_{(1)}$-$S_{(1)pred}$ time series plotted in Figure 4

is 2.6 hr, similar to the Figure 1 autocorrelation time of the full 1991-2007 time series of $S_{(1)}$-$S_{(1)pred}$.

Additionally, it would be valuable to differentiate $S_{(1)}$ from other indices commonly used to characterize magnetospheric activity. In order to achieve this task we use the methodology of Rosso et al. (2007) based on the combined use of permutation entropy (Bandt and Pompe, 2002) and Jensen-Shannon complexity mapping. This mapping developed by Rosso et al. (2007) is particularly useful to disentangle deterministic and stochastic time series. The reader with little

familiarity to these two information theoretic measures can consult the reviews of Riedl et al. (2013) and Zanin et al. (2012) or the pedestrian methodology section found in Osmane et al. (2019). In Figures 5 and 6, we map the values of AL (red), am (blue), $S_{(1)}$ (black) and $D_{(1)}$ (pink) on the complexity-entropy plane for an interval of 500 hours duration with no data gaps, embedding dimensions of d=4 and embedding delay T ranging between 2 hours and 40 hours. Because there are a total of



d!=24 ordinal patterns we are limited to embedding delays of the order of two days. For embedding delays greater than T=48 hours the number of segments N-(d-1)*T becomes too small to ascertain the likelihood of forbidden ordinal patterns. Error bars for the Jensen-Shannon complexity, shown for the zoomed panels of the complexity-entropy planes, are computed as the square root of the number of ordinal patterns divided by the number of segments available. Hence, larger embedded

dimensions d, require larger number of segments N-(d-1)*T to determine whether the Jensen-Shannon complexity lies significantly above the stochastic boundary (see below).

The bottom left panels in Figures 5 and 6 show the value of the permutation entropy for AL, am, $D_{(1)}$ and $S_{(1)}$ as a function of embedding delay. Similarly, the bottom right panels show the value of the Jensen-Shannon complexity for AL, am, $D_{(1)}$ and $S_{(1)}$ as a function of embedding delay. What we notice is that all four signals are highly stochastic since the

normalized permutation entropy is very close to 1. However, we see that the Jensen-Shannon complexity for $S_{(1)}$ is of comparable magnitude as for am, and that it is significantly larger than for AL. This is not a surprise because the construction of $S_{(1)}$ was based on am, and the Jensen-Shannon complexity is indicating that the former preserved the correlated structures of the latter on timescales ranging between a few hours to a few days. The top left panel of Figures 5 and 6 shows the complexity-entropy plane and the top right panel is a zoom of the right corner where most of the data for

AL, am, and $S_{(1)}$ is lying. On both figures the blue line curves represent the maximum and minimum value of complexity for a fixed entropy value, and the dashed curve represents the complexity-entropy mapping of fractional Brownian motion (fBm) with Hurst exponent ranging between 0 and 1, that is a stochastic process that also contains correlated structures. The fBm curve is a boundary between deterministic (above) and stochastic (below) fluctuations. We note that AL is effectively stochastic, whereas am and $S_{(1)}$ lie above the fBm boundary for a few tens of hours. The explanation for this behavior from

am lies from its construction: it is repeated for three hours at a time. Hence, ordinal patterns of size d=4 and embedding delays of a few hours will register the repetition as correlated structures. Since $S_{(1)}$ is constructed in part with am, it also contains part of its correlated structure.

For longer embedding delays of the order of seasonal variations ranging from 27 to 45 days (not shown), all four time series overlap and are indistinguishable from stochastic fluctuations. In terms of complexity-entropy plane it translates

into a permutation entropy of approximately 1 and a Jensen-Shannon complexity of almost 0. Hence, the system variable $S_{(1)}$, based on various magnetospheric indices, preserves the stochastic and correlational structures of its individual components.

### 3.2. The Secondary Modes of Reaction Represented by $S_{(2)}$ and $S_{(3)}$.

In Figures 7a and 7b the second and third scalar pairs are plotted, $S_{(2)}$ as a function of $D_{(2)}$ and $S_{(3)}$ as a function of $D_{(3)}$, respectively. (See expressions (2) and (3).) The correlation coefficient for the second pair is still quite high ($r_{corr} = 0.775$), but lower than that of the first pair (Figure 1). This correlation coefficient $r_{corr} = 0.775$ for the secondary mode is better than correlations obtained in most studies of solar-wind/magnetosphere coupling using single measures of the magnetospheric system (e.g. Table 3 of Newell et al., 2007; Table 1 of Borovsky, 2013). $D_{(2)}$ describes $r_{corr}^2 = 60.0\%$ of the variance of $S_{(2)}$.

In Figure 7b the correlation coefficient for the third pair is low ($r_{corr} = 0.456$); $D_{(3)}$ only describes $r_{corr}^2 = 20.8\%$ of the variance of $S_{(3)}$. Canonical pairs beyond the third pair have even weaker correlations.

Figure 2c shows that mode $S_{(2)}$ (Figure 7a) is dominated by opposite-signed coefficients for $mP_i$ and $mP_e$, which respectively are measures of the global ion precipitation into the atmosphere versus the global electron precipitation into the atmosphere. In this $S_{(2)}$ mode the intensity of ion and electron precipitation reacts oppositely. Figure 2d shows that $D_{(2)}$ (the

driver of $S_{(2)}$) is dominated by the solar wind number density $n_{sw}$ opposite to the clock angle $\sin^2(\theta_{clock}/2)$ of the solar-wind magnetic field, with the solar wind density increasing while the clock angle decreases resulting in more ion precipitation and less electron precipitation. This ion-versus-electron precipitation mode is a newly uncovered mode of reaction of the M-I-T system to the solar wind.





Figure 2e shows that $S_{(3)}$ (Figure 7b) is characterized by PCI and $mP_i$ acting oppositely to Kp and am. PCI is a measure of high-latitude electrical currents in the magnetosphere and $mP_i$ is a measure of high-latitude ion precipitation; Kp and am are measures of global magnetospheric convection. This $S_{(3)}$ mode is very similar to a high-latitude versus convection mode uncovered by Borovsky (2014) and by Holappa et al. (2014). Figure 2f indicates that the driver $D_{(3)}$ for this

5   mode is the solar wind velocity acting oppositely to the magnetic field clock angle: the wind velocity increasing while the clock angle is reduced producing more convection and less high-latitude activity, or the wind slowing down while the clock angle increases producing less convection and more high-latitude activity.

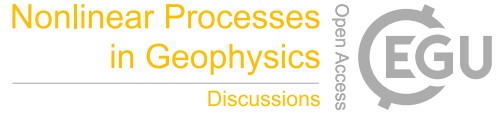

**4. Advantages of the Reduced (Aggregate-Variable) Representation of the System**

The aggregate variable $S_{(1)}$ acts as a global activity index for the magnetospheric system: $S_{(1)}$ is new and unfamiliar and experience using $S_{(1)}$ is needed to gain an understanding of the full usefulness of this measure. $S_{(1)}$ could be thought of as a next-generation magnetospheric index. In Earth systems science global aggregate variables are familiar: the Global

Warming Index (Hasselman, 1997; Haustein et al., 2016), the global mean sea level (Vermeer and Rahmstorf, 2009), the mean global temperature (Hansen et al., 2006), the Palmer Drought Severity Index (Wells et al., 2004), and Sea Surface Temperature indices (Kaplan et al., 1998). are well known. in Earth systems science aggregate variables such as Sea Surface Temperature indices (Kaplan et al., 1998), the Global Warming Index (Haustein et al., 2016), and the Palmer Drought Severity Index (Wells et al., 2004). Here the aggregate variable $S_{(1)}$ is mathematically derived. The individual variables of $\underline{S}$

that go into the definition of $S_{(1)}$ represent familiar and identifiable aspects of activity in the magnetospheric system. The composite variable $S_{(1)}$ is a mix of these understood measurements, the mix reflecting some global properties of the system's reaction to the solar wind. Unfamiliar as it is, the composite-scalar $D_{(1)} \rightarrow S_{(1)}$ reduction of the state-vector $\underline{D} \rightarrow \underline{S}$ picture exhibits some outright advantages for the magnetospheric system. This is particularly true in comparison with the standard method of analysis of magnetospheric driving by the solar wind that uses only a single measurement of magnetospheric

activity and a single function of solar-wind variables. Four advantages are discussed in the following four paragraphs.

*Linearity*. The plotted points in Figure 1 indicate that there is a linear response of the composite system variable $S_{(1)}(t)$ to the composite driver $D_{(1)}(t)$. Usually, single measures of the magnetosphere tend to have a nonlinear response to the solar wind (e.g. Voros, 1994; Valdivia et al., 1996; Sharma et al., 2005b; Borovsky, 2013; Stepanova and Valdivia, 2016), with the individual activity variables saturating (becoming anomalously weak) when solar-wind driving becomes strong (e.g.

Fig. 3 of Reiff and Luhmann, 1986; Fig. 17 of Lavraud and Borovsky, 2008; Fig. 6 of Borovsky, 2013). Such a saturation is not seen for $S_{(1)}$ driven by $D_{(1)}$. Undoubtedly, the linearity of the result is in part owed to the maximizing of the "linear" correlation coefficient in the CCA process. The linearity of the $S_{(1)}$-versus-$D_{(1)}$ relation has a great advantage: the same mathematical relationship between $S_{(1)}$ and $D_{(1)}$ (i.e. expression (4)) holds for weak driving of the system (small $D_{(1)}$) (e.g. Kerns and Gussenhoven, 1990) and for strong driving of the system (large $D_{(1)}$) (e.g. Sharma and Veeramani, 2011).

*Low Noise*. The high correlation between $S_{(1)}$ and $D_{(1)}$ (cf. Figure 1) indicates that there is a relatively low level of noise in the linear-regression fit to $S_{(1)}$: the activity in the system as described by $S_{(1)}$ responds directly to the solar-wind driving as described by $D_{(1)}$. For example, the unaccounted for variance of $S_{(1)}$ is only 15.2%. Single measures of the magnetospheric system have much weaker Pearson linear correlation coefficients with solar-wind variables than do $S_{(1)}$ and $D_{(1)}$. Examples can be found in Table 3 of Newell et al. (2007) and Table 1 of Borovsky (2013): the maximum correlation

coefficient in those tables is 0.860 (for the Dst index), but usually it is much lower. The lower noise is also confirmed by the Jensen-Shannon complexity analysis of $S_{(1)}$: the points for $S_{(1)}$ and $D_{(1)}$ sit closer to the maximum complexity curve than AL and other indices. The lower noise (and higher $r_{corr}$) reduces "regression dilution bias" (Bock and Petersen, 1975; Hutcheon et al., 2010) when the system activity is fit by the driver strength. Regression dilution bias can lead to spurious interpretation of trends in the data when subsets of the data are compared, particularly when a subset with systematically weaker driving is

compared with a subset with systematically stronger driving.

*High Prediction Efficiency*. In magnetospheric physics, predicting what the reaction of the magnetospheric system will be to measured upstream solar-wind conditions is very important: i.e. the prediction of "space weather" (Singer et al., 2001). The high correlation between $S_{(1)}$ and $D_{(1)}$ means that there will be a high prediction efficiency when the value of $S_{(1)}$ is predicted from a knowledge of the value of $D_{(1)}$. Note that this is high prediction of $S_{(1)}(t)$ without using past values of $S_{(1)}$,

just using the present value of $D_{(1)}(t)$. By optimizing the Pearson linear correlation coefficient between $\underline{S}$ and $\underline{D}$, $S_{(1)}$ was designed to focus on aspects of the magnetospheric system that are responsive to the conditions of the solar wind. Internal dynamics of the system that are not dependent on the time-varying state of the driver are de-emphasized in $S_{(1)}$.





*Compactness of the Description*. Reductionist analysis has concluded that the magnetosphere-ionosphere-thermosphere system is extremely complicated (e.g. Siscoe, 2011; Eastwood et al., 2015; Borovsky and Valdivia, 2018) and as driven by the solar wind there are major outstanding issues as to how the system functions (e.g. Denton et al., 2016). Having a single scalar variable $S_{(1)}(t)$ that is describing a universal global reaction of the system to its driver promises to

yield insight as to how the combined system operates.

*Uncovering New Modes of Reaction*. In the CCA analysis of the system and driver state vectors, two additional aggregate variables $S_{(2)}(t)$ and $S_{(3)}(t)$ were generated (expressions (2a) and (3a)). Analysis in Section 3.B showed these two variables to represent two modes of reaction of the system to the driver that are independent of (uncorrelated with) the global-activity mode represented by $S_{(1)}(t)$. The mode represented by $S_{(3)}$ is known (having been independently discovered

by this CCA methodology in Borovsky (2014) and by a principle-components methodology in Holappa et al. (2014)), but the mode represented by $S_{(2)}$ has until now been unknown. The CCA methodology used here also identifies the aggregate driver variable that drives each of the independent modes. In future, expanding the system state vector to include a larger number of measurements in the diverse magnetospheric system should enable this state-vector-reduction methodology to uncover more unknown modes of reaction of the system to the driver. Once a reaction and its driver are uncovered, reductionist

analysis can be applied to determine the physical reasons why the mode arises.

For a system measured by multiple time-dependent variables (that are collected into a time-dependent system state vector $\underline{S}(t)$), with that system driven by multiple time-dependent factors (inputs) (that are collected into a time-dependent driver state vector $\underline{D}(t)$), canonical correlation analysis (CCA) can be used to reduce the $\underline{D}(t) \rightarrow \underline{S}(t)$ state-vector picture to a $D_{(i)}(t) \rightarrow S_{(i)}(t)$ composite-scalar picture. The reduction will work, even if there is influence on the driver by the system (i.e.

$\underline{D}(t) \leftrightarrow \underline{S}(t)$). The advantageous properties of this reduction that were examined for the magnetospheric system should apply to systems in general.

Future developments of this methodology will focus on the introduction of time lags between the driver and the system, on the introduction of integro-differential correlations rather than algebraic correlations (e.g. Borovsky, 2017), and on the use of dynamic canonical correlation analysis (e.g. Dong and Qin, 2018a,b).


**Appendix: The Variables Comprising the Magnetospheric System State Vector and the Solar-Wind Driver State Vector**

The time-dependent variables of the magnetospheric system state vector and the solar-wind driver state vector are listed in Table 1.

The magnetospheric variables measure various aspects of activity in the magnetosphere. The auroral upper index AU (Davis and Sugiura, 1966) measures electrical current in the high-latitude ionosphere: this variable is taken to be a measure of electrical currents in the dayside magnetosphere (Goertz et al., 1993). The auroral lower index AL (Davis and Sugiura, 1966) measures electrical current in the high-latitude nightside ionosphere: this variable is taken to be a measure of auroral activity in the nightside magnetosphere (Goertz et al., 1993). The polar cap index PCI is a measure of the strength of

cross-polar-cap electrical current in the ionosphere (Troshichev et al., 1988). The planetary K index Kp is a measure of the strength of global convection in the magnetosphere (Thomsen, 2004). The range index am (Mayaud, 1980) is another measure of the strength of global magnetospheric convection. The disturbance storm-time index Dst measures plasma pressure in the inner magnetosphere (Dessler and Parker, 1959); Dst also reacts to the currents on the dayside boundary of the magnetosphere and to the cross-magnetotail currents in the nightside magnetosphere. The time derivative of the

magnitude of the Dst index d|Dst|/dt is a compound measure of magnetospheric activity: when d|Dst|/dt is positive, hot plasma is being convected from the magnetotail into the dipolar portion of the magnetosphere, and when d|Dst|/dt is negative, convection has recently subsided. The variables $mP_e$ and $mP_i$ are estimates of the full-Earth power in magnetospheric electron precipitation into the atmosphere and magnetospheric ion precipitation into the atmosphere (Emery





et al., 2008, 2009), with these estimates coming from observations on only a few spacecraft in orbit around the Earth. The average of the ion-plasma-sheet particle pressure $P_{ips}$ around the Earth (Borovsky, 2017) is obtained from 3 to 5 spacecraft.

The variables going into the solar-wind driver state vector are various measures of the time-dependent solar wind at Earth. The solar-wind speed $v_{sw}$ ranges from 244 km/s to 1045 km/s in the 1991-2007 data set. The solar-wind number

density ranges from 0.3 particles/cm$^3$ to 98.2 particles/cm$^3$ in the data set. $B_z$ is the magnetic-field component in the solar-wind plasma that is approximately aligned with the Earth's magnetic-dipole orientation. The function f(M) (Borovsky and Birn, 2014) is a function of the solar-wind Mach number M that accounts for the properties of the bow shock that forms upstream of the Earth in the supersonic solar-wind flow. The clock angle $\theta_{clock}$ measures the angular alignment of the solar-wind magnetic-field vector with the Earth's magnetic-dipole orientation. The angle $\theta_{Bn}$ measures the orientation of the solar-

wind magnetic-field vector with respect to the Sun-Earth line. $F_{10.7}$ is the 10.7-cm radio flux from the Sun, a proxy for the ionization of the upper atmosphere of the Earth by solar photons.

*Data availability.* The 1991-2007 data set of hourly values of $S_{(1)}$, $S_{(2)}$, $S_{(3)}$, $D_{(1)}$, $D_{(2)}$, and $D_{(3)}$ has been made available at DOI 10.5281/zenodo.1560686 and at DOI 10.17605/OSF.IO/QYTNJ as a tab-delimited text file.

*Author contributions.* JEB devised this study and performed the CCA analysis. AO performed the complexity and entropy analysis. Both authors are responsible for the interpretation of the results and for the writing of the manuscript.

*Competing interests.* The authors declare that they have no conflict of interest.

**Acknowledgements.** The authors thank Mick Denton and Juan Alejandro Valdivia for helpful discussions. This work was

supported by the NSF GEM Program via award AGS-1502947, by the NASA Heliophysics LWS TRT program via grants NNX16AB75G and NNX14AN90G, by the NSF Solar-Terrestrial Program via grant AGS-12GG13659, by the NASA Heliophysics Guest Investigator Program via grants NNX17AB71G, by the NSF SHINE program via award AGS-1723416 and by the Academy of Finland via grant #297688/2015.


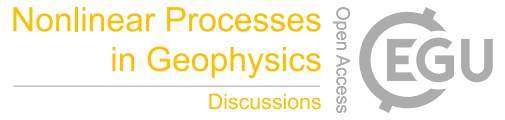



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





Table 1. The 9 time-dependent variables going into the system state vector $\underline{S}(t)$ of the magnetosphere and the 8 time-dependent variables going into the driver state vector $\underline{D}(t)$ of the solar wind.

| System (Magnetospheric) Variables | Driver (Solar-Wind) Variables |
|---|---|
| Auroral Lower index AL | Solar wind speed $v_{sw}$ |
| Auroral Upper index AU | Solar wind number density $n_{sw}$ |
| Polar Cap Index PCI | Solar 10.7-cm radio flux $F_{10.7}$ |
| Planetary K index Kp | North-south-component magnetic field $-B_z$ |
| Geomagnetic range index am | Mach-number function $f(M)$ |
| Time derivative of disturbance storm-time index Dst | Magnetic-field clock angle $\theta_{clock}$ |
| Hemispheric electron precipitation power $mP_e$ | Magnetic-field Sun-Earth angle $\theta_{Bn}$ |
| Hemispheric ion precipitation power $mP_i$ | Magnetic-field-vector fluctuation amplitude $|\Delta\underline{B}|$ |
| Pressure of the ion plasma sheet $P_{ips}$ | |




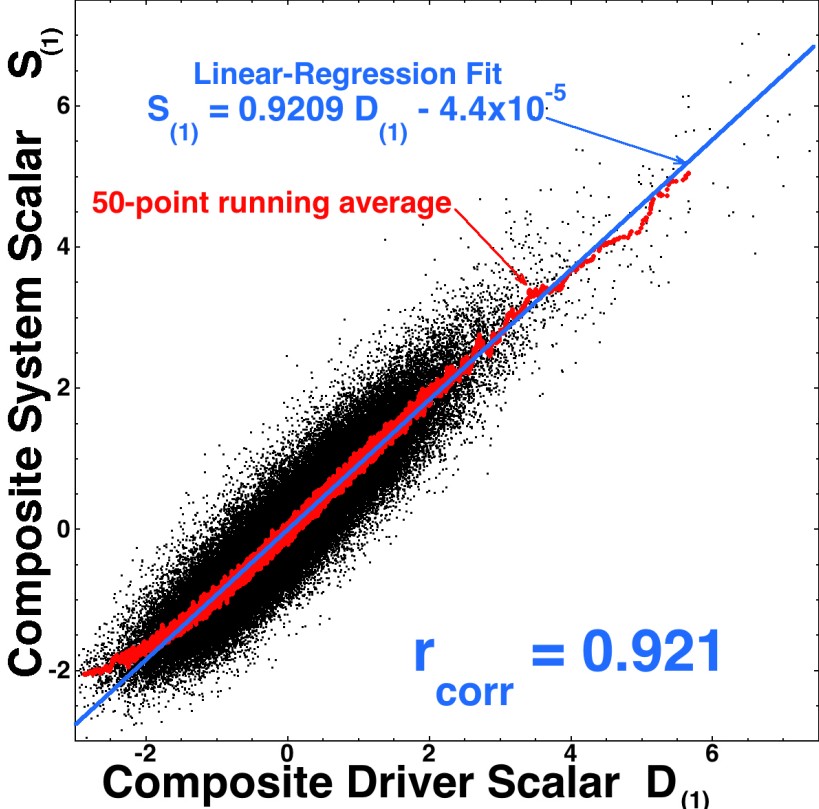

Figure 1. The aggregate system scalar $S_{(1)}$ is plotted as a function of the driver scalar $D_{(1)}$ for the 1-hr-resolution 1991-2007

5    data set. Each black point is 1 hour of data.



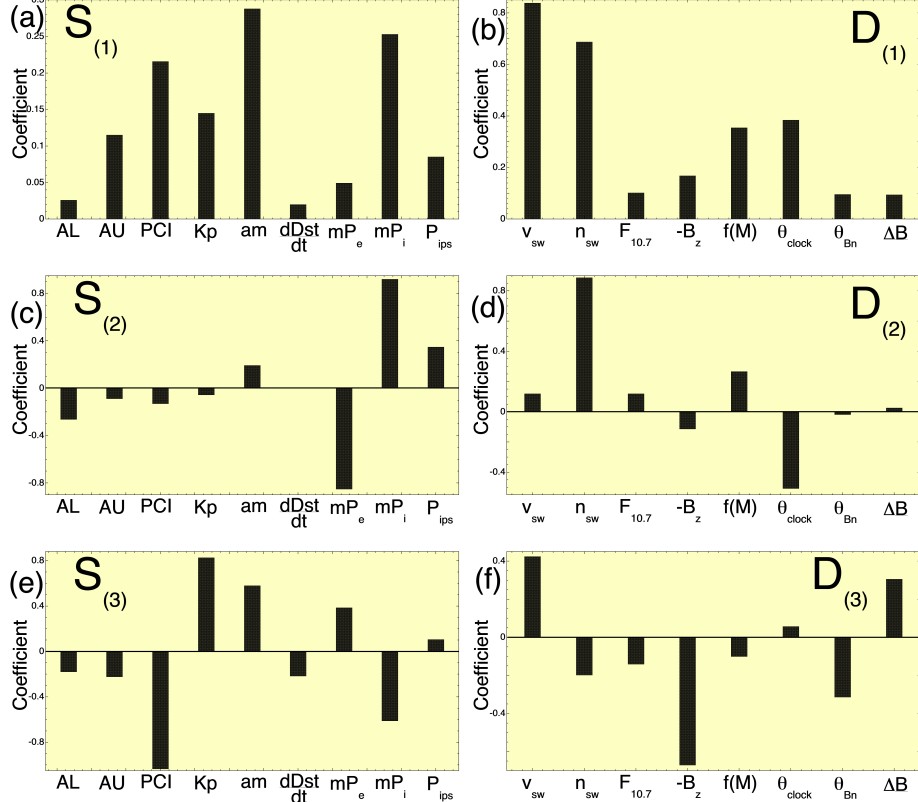

Figure 2. Plots of the 9 components of the coefficient vectors used to project the system state vector $\underline{S}$ into the aggregate variables $S_{(1)}$ (panel a), $S_{(2)}$ (panel c), and $S_{(3)}$ (panel e) and plots of the 8 components of the coefficient vectors used to project the driver state vector $\underline{D}$ into the driver scalar variables $D_{(1)}$ (panel a), $D_{(2)}$ (panel c), and $D_{(3)}$ (panel e).



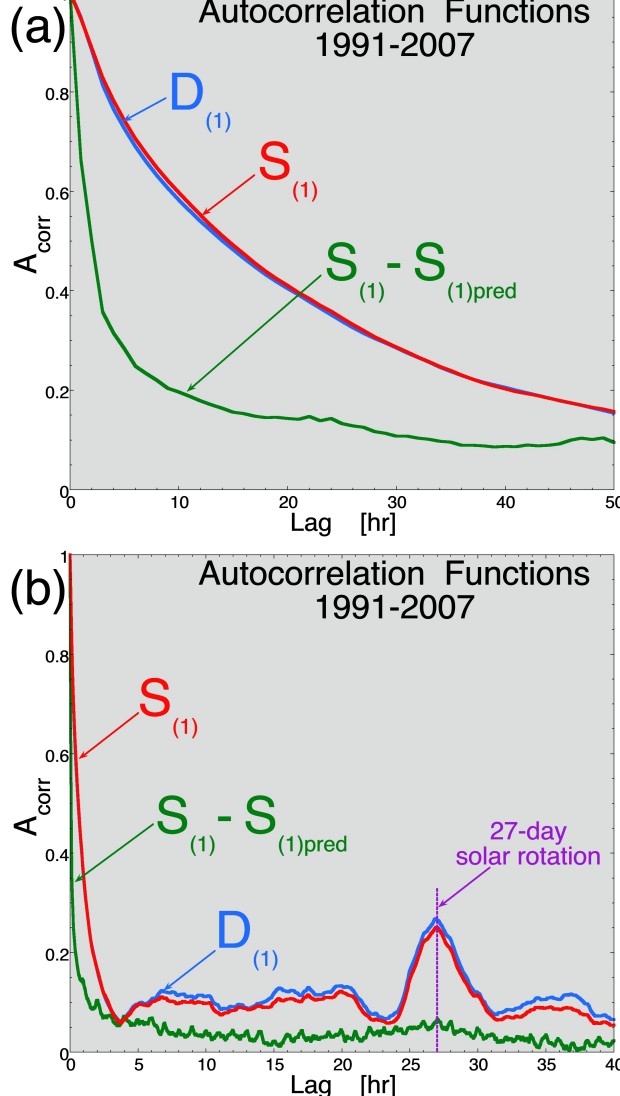

Figure 3. The autocorrelation functions for the system scalar $S_{(1)}$ (red), the driver scalar $D_{(1)}$ (blue), and the unaccounted for variance $E_{(1)}-E_{(1)pred}$ (green) are plotted. In panel (a) the plot extends to 50 hours and in panel (b) the plot extends to 40 days.




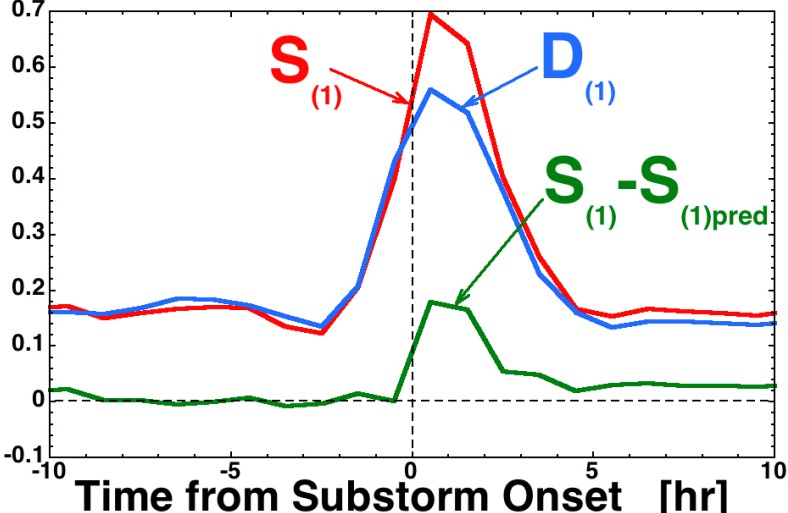

Figure 4. Superposed epoch averages of $S_{(1)}$ (red), $D_{(1)}$ (blue), and $S_{(1)}-S_{(1)pred}$ (green) for 2155 substorms. The epoch time ($t=0$) is the time of onset of each substorm.



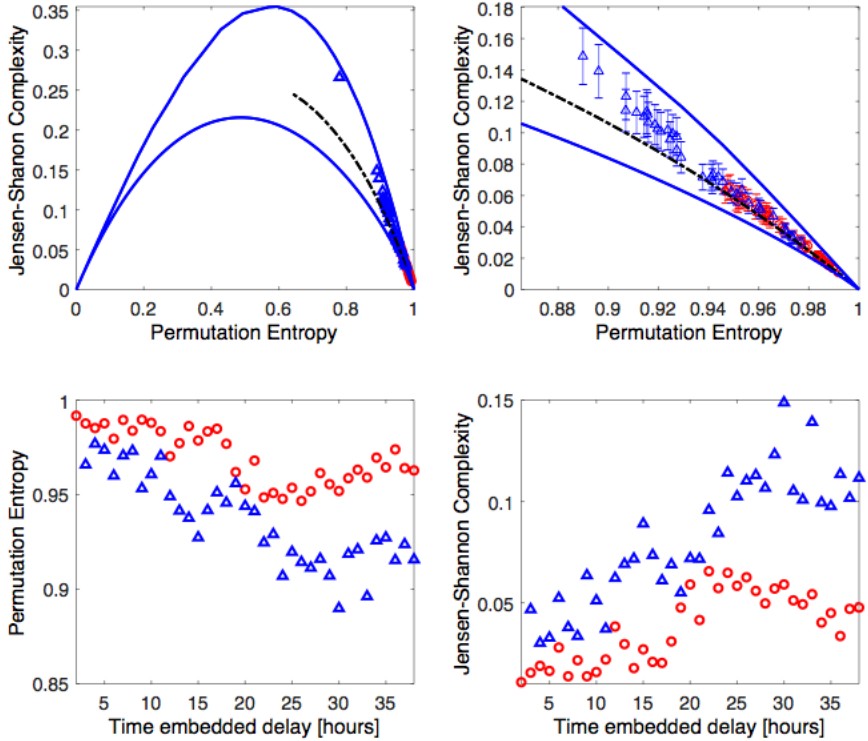

Figure 5. Jensen-Shannon complexity and permutation entropy for $S_{(1)}$ (black square), AL (red) and am (blue) for 500 points with no gaps sampled at 1 hour interval, embedding dimension d=4 and embedding delays ranging between 2 hours and 40 hours.

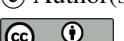



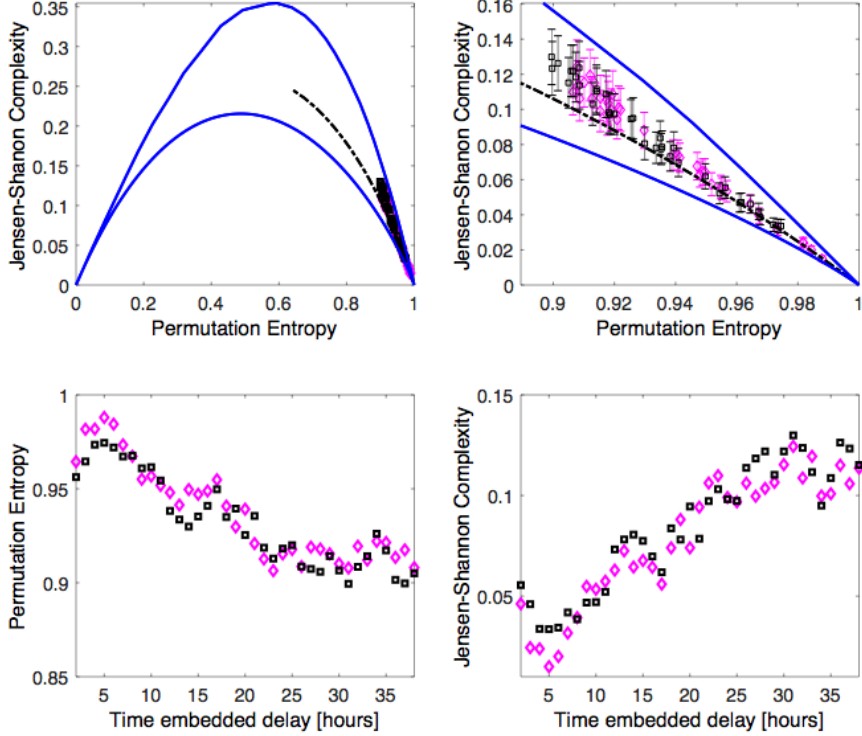

Figure 6. Jensen-Shannon complexity and permutation entropy for $S_{(1)}$ (black), $D_{(1)}$ (pink) for 500 points with no gaps sampled at 1 hour interval, embedding dimension d=4 and embedding delays ranging between 2 hours and 40 hours.

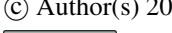


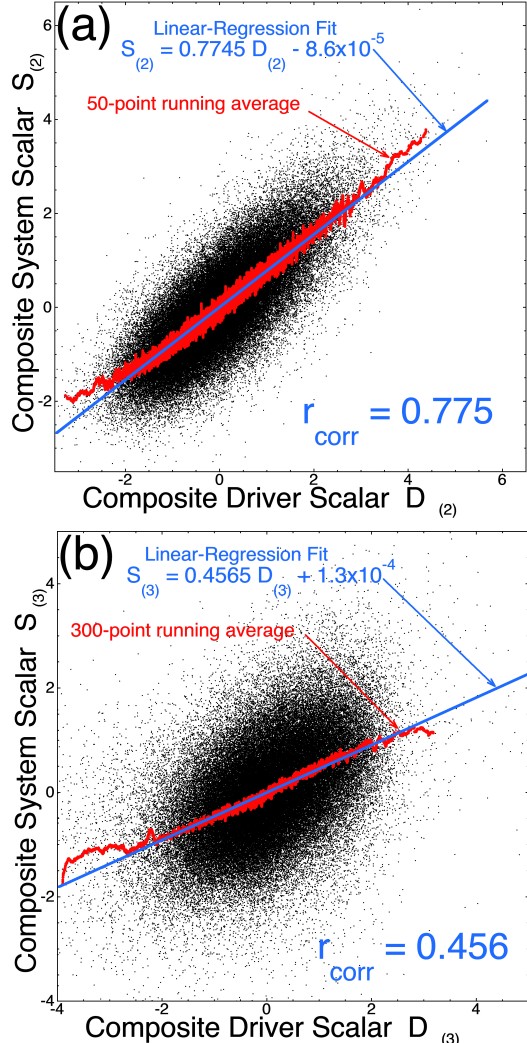

Figure 7. For the 1-hr-resolution 1991-2007 data set, the aggregate system scalar $S_{(2)}$ is plotted as a function of the driver scalar $D_{(2)}$ in panel (a) and the aggregate system scalar $S_{(3)}$ is plotted as a function of the driver scalar $D_{(3)}$ in panel (b). Each black point is 1 hour of data.

