# Peer review of "Compacting the Description of a Time-Dependent Multivariable System and Its Multivariable Driver by Reducing the State Vectors to Aggregate Scalars: The Earth's Solar-Wind-Driven Magnetosphere"

_Nonlinear Processes in Geophysics, 2019_

## Referee Comment (RC1) · Anonymous Referee #1 · 25 Jun 2019

General comments

This is an interesting paper demonstrating an application of the canonical correlation analysis technique to magnetospheric state variables. The canonical correlation analysis produces three leading canonical variables representing different physical processes in the magnetosphere. In addition to the CCA method the authors use Jensen-

Shannon complexity and permutation entropy analysis methods. In my opinion, the results and implications of canonical correlation analysis are clearly presented, but the manuscript lacks a proper discussion on the implications of Jensen-Shannon complexity and permutation entropy analysis results. I think that the main purpose of this paper is to demonstrate the combined CCA / complexity analysis methodology to a wide audience of geophysicists. Thus, the paper would be much stronger if the authors could explain in detail what value does the complexity analysis add on top of the CCA analysis. I suggest that the authors add a concise review on the complexity analysis methods and explain their implications better.

Specific comments

The authors state on line 25 of page 7 that "the system variable S1, based on various magnetospheric indices, preserves the stochastic and correlational structures of its individual components". I do not understand the significance of this result. Is the preservation of the correlational structures in linear combinations (S1-S3) surprising? Would the opposite result be even possible? The CCA analysis reveals that geomagnetic indices (such as PC and Kp) respond to differently to different solar wind parameters which gives rise to the third canonical variable, verifying earlier results by Borovsky et al. (2014) and Holappa et al. (2014). Interestingly, the analysis also reveals a new independent mode arising from the difference in the electron and ion precipitation power. The physics behind this result is (understandably) not discussed in this paper as it will require more detailed data analysis. However, I suggest that the authors highlight this result in the abstract. Line 8: What would be the correlation coefficient between S1 and the best fitting solar wind coupling function? Is it significantly worse than the correlation between S1 and D1?

---

## Referee Comment (RC2) · Marina Stepanova (Referee) · 26 Jun 2019

The authors analyze the solar-wind-driven magnetosphere-ionosphere-thermosphere system using a state-vector description of a time-dependent driven system. They develop a methodology based on the canonical correlation analysis to reduce the

time-dependent system and driver state vectors to time-dependent system and driver scalars. This allows them to find the scalars which describe the response in the system that is most-closely related to the driver, obtaining the following advantages: low noise, high prediction efficiency, linearity in the described system response to the driver, and compactness. The methodology also identifies independent modes of reaction of a system to its driver, and to assess the properties of the derived aggregate scalars using autocorrelation analysis, Jensen-Shannon complexity analysis, and permutation-entropy analysis.

In particular, it was shown that the aggregate variable S(1) can be used as a global activity index for the magnetospheric system, i.e. a next-generation magnetospheric index which would play a role similar to the Global Warming Index, and other global aggregate variables in the Earth system science.

Without doubts, this approach developed by Joseph Borovsky and co-authors during last few years is very novel and promising, and the paper reflects current advances in the improvement of the analysis of the dynamics of the magnetosphere represented as a state vector and should be published in the Nonlinear Processes in Geophysics after minor revision.

Minor points:

The title of the article is too long.

I would switch the second and the first paragraph of the introduction, starting with the description of the magnetosphere, and may be adding a few examples of the use of large data series for the prediction of the properties of the magnetosphere in the past and its limitations, justifying in this way the necessity to use a new methodology based on the state-vector approach and the canonical correlation analysis, described in the first paragraph.

With best wishes

**NPGD**

Marina Stepanova

Interactive
comment

---

## Author Comment (AC2) · 16 Aug 2019

The authors have added text to the paper to better motivate the reason the complexity analysis was performed.

Concerning the reviewers comments about page 25:The preservation of the correla-

tional structures in the linear combinations S1-S3 is not surprising, but not guaranteed either. The permutation entropy is invariant under any monotonic transformations (for instance, if one scales the time series by a positive real number, or if one were to take the logarithm). However, if one used a linear combinations of non-monotonic functions, for instance some linear combination of trigonometric function, then the permutation entropy would not be invariant. Since the Jensen-Shannon complexity is a function of the permutation entropy, it is also invariant under monotonic transformations. Additionally, if one takes an average around the mean of some time series over a time Tau, one will reduce the noise level for fluctuations with timescales less or comparable to Tau. Thus, the stochastic nature of the signal will be reduced, and the permutation entropy and Jensen-Shannon complexity would move up in the plane towards the chaotic or/and periodic regions.

We have clarified this point in the text on page 7 and 8 and have added a few sentences to justify the use of the Jensen-Shannon complexity on page 7 as well.

Please also note the supplement to this comment:
https://www.nonlin-processes-geophys-discuss.net/npg-2019-2/npg-2019-2-AC2-supplement.pdf

---

## Author Response (AR1)

General comments

The authors thank the reviewer for the constructive comments.

This is an interesting paper demonstrating an application of the canonical correlation analysis technique to magnetospheric state variables. The canonical correlation analysis produces three leading canonical variables representing different physical pro- cesses in the magnetosphere. In addition to the CCA method the authors use Jensen-Shannon complexity and permutation entropy analysis methods. In my opinion, the results and implications of canonical correlation analysis are clearly presented, but the manuscript lacks a proper discussion on the implications of Jensen-Shannon complexity and permutation entropy analysis results. I think that the main purpose of this paper is to demonstrate the combined CCA / complexity analysis methodology to a wide audience of geophysicists. Thus, the paper would be much stronger if the authors could explain in detail what value does the complexity analysis add on top of the CCA analysis. I suggest that the authors add a concise review on the complexity analysis methods and explain their implications better.

The authors have added text to the paper to better motivate the reason the complexity analysis was performed.

Specific comments

The authors state on line 25 of page 7 that "the system variable S1, based on various magnetospheric indices, preserves the stochastic and correlational structures of its individual components". I do not understand the significance of this result. Is the preservation of the correlational structures in linear combinations (S1-S3) surprising? Would the opposite result be even possible?

Concerning the reviewers comments about page 25:The preservation of the correlational structures in the linear combinations S1-S3 is not surprising, but not guaranteed either. The permutation entropy is invariant under any monotonic transformations (for instance, if one scales the time series by a positive real number, or if one were to take the logarithm). However, if one used a linear combinations of non-monotonic functions, for instance some linear combination of trigonometric function, then the permutation entropy would not be invariant. Since the Jensen-Shannon complexity is a function of the permutation entropy, it is also invariant under monotonic transformations. Addition- ally, if one takes an average around the mean of some time series over a time Tau, one will reduce the noise level for fluctuations with timescales less or comparable to Tau. Thus, the stochastic nature of the signal will be reduced, and the permutation entropy and Jensen-Shannon complexity would move up in the plane towards the chaotic or/and periodic regions.

We have clarified this point in the text on page 7 and 8 and have added a few sentences to justify the use of the Jensen-Shannon complexity on page 7 as well .

The CCA analysis reveals that geomagnetic indices (such as PC and Kp) respond to differently to different solar wind parameters which gives rise to the third canonical variable, verifying earlier results by Borovsky et al. (2014) and Holappa et al. (2014). Interestingly, the analysis also reveals a new in- dependent mode arising from the difference in the electron and ion precipitation power. The physics behind this result is (understandably) not discussed in this paper as it will require more detailed data analysis. However, I suggest that the authors highlight this result in the abstract.

As suggested, this finding is now stated in the abstract.

Line 8: What would be the correlation coefficient between S1 and the best fitting solar wind coupling function? Is it significantly worse than the correlation between S1 and D1?

To address this question, on page 5 of the manuscript a new paragraph has been added that compares the correlations of S(1) with several coupling functions. The correlations are significantly worse than the correlation with D(1).

Marina Stepanova (Referee)
marina.stepanova@usach.cl

Thank you Marina for your comments.

The authors analyze the solar-wind-driven magnetosphere-ionosphere-thermosphere system using a state-vector description of a time-dependent driven system. They develop a methodology based on the canonical correlation analysis to reduce the time-dependent system and driver state vectors to time-dependent system and driver scalars. This allows them to find the scalars which describe the response in the system that is most-closely related to the driver, obtaining the following advantages: low noise, high prediction efficiency, linearity in the described system response to the driver, and compactness. The methodology also identifies independent modes of reaction of a system to its driver, and to assess the properties of the derived aggregate scalars using autocorrelation analysis, Jensen-Shannon complexity analysis, and permutation- entropy analysis.

In particular, it was shown that the aggregate variable $S(1)$ can be used as a global activity index for the magnetospheric system, i.e. a next-generation magnetospheric index which would play a role similar to the Global Warming Index, and other global aggregate variables in the Earth system science.

Without doubts, this approach developed by Joseph Borovsky and co-authors during last few years is very novel and promising, and the paper reflects current advances in the improvement of the analysis of the dynamics of the magnetosphere represented as a state vector and should be published in the Nonlinear Processes in Geophysics after minor revision.

Minor points:

The title of the article is too long.

As suggested, the title of the article has been shortened.

I would switch the second and the first paragraph of the introduction, starting with the description of the magnetosphere, and may be adding a few examples of the use of large data series for the prediction of the properties of the magnetosphere in the past and its limitations, justifying in this way the necessity to use a new methodology based on the state-vector approach and the canonical correlation analysis, described in the first paragraph.

As suggested, the order of the two paragraphs has been switched.

With best wishes

Marina Stepanova